# Novel Self-Forming Nanosized DDS Particles for BNCT: Utilizing A Hydrophobic Boron Cluster and Its Molecular Glue Effect

**DOI:** 10.3390/cells11203307

**Published:** 2022-10-21

**Authors:** Abdul Basith Fithroni, Kazuko Kobayashi, Hirotaka Uji, Manabu Ishimoto, Masaru Akehi, Takashi Ohtsuki, Eiji Matsuura

**Affiliations:** 1Department of Interdisciplinary Science and Engineering in Health Systems, Okayama University, Okayama 700-8530, Japan; 2Department of Cell Chemistry, Graduate School of Medicine, Dentistry, and Pharmaceutical Sciences, Okayama University, Okayama 700-8558, Japan; 3Collaborative Research Center for OMIC, Graduate School of Medicine, Dentistry, and Pharmaceutical Sciences, Okayama University, Okayama 700-8558, Japan; 4Department of Material Chemistry, Graduate School of Engineering, Kyoto University, Kyoto 615-8510, Japan; 5Fukushima SiC Applied Engineering Inc., Naraha-machi 979-0513, Japan; 6Neutron Therapy Research Center (NTRC), Okayama University, 2-5-1 Shikata-cho, Kita-ku, Okayama 700-8558, Japan

**Keywords:** boron neutron capture therapy (BNCT), biologically self-degradable amphipathic polymer (Lactosome), hydrophobic boron cluster, carborane isomers or *o*-carborane alkylated derivatives, molecular glue effect

## Abstract

BNCT is a non-invasive cancer therapy that allows for cancer cell death without harming adjacent cells. However, the application is limited, owing to the challenges of working with clinically approved boron (B) compounds and drug delivery systems (DDS). To address the issues, we developed self-forming nanoparticles consisting of a biodegradable polymer, namely, “AB-type Lactosome (AB-Lac)” loaded with B compounds. Three carborane isomers (*o-*, *m-*, and *p-*carborane) and three related alkylated derivatives, i.e., 1,2-dimethy-*o-*carborane (diC1-Carb), 1,2-dihexyl-*o-*carborane (diC6-Carb), and 1,2-didodecyl-*o-*carborane (diC12-Carb), were separately loaded. diC6-Carb was highly loaded with AB-Lac particles, and their stability indicated the “molecular glue” effect. The efficiency of in vitro B uptake of diC6-Carb for BNCT was confirmed at non-cytotoxic concentration in several cancer cell lines. In vivo/ex vivo biodistribution studies indicated that the AB-Lac particles were remarkably accumulated within 72 h post-injection in the tumor lesions of mice bearing syngeneic breast cancer (4T1) cells, but the maximum accumulation was reached at 12 h. In ex vivo B biodistribution, the ratios of tumor/normal tissue (T/N) and tumor/blood (T/Bl) of the diC6-Carb-loaded particles remained stably high up to 72 h. Therefore, we propose the diC6-Carb-loaded AB-Lac particles as a promising candidate medicine for BNCT.

## 1. Introduction

Boron (B) neutron capture therapy (BNCT) is based on the nuclear capture and fission reactions that occur once the stable (but not radioactive) isotope B-10 (^10^B) is irradiated with either low-energy (0.025 eV) thermal neutrons or epithermal neutrons (10,000 eV) that become thermalized as they penetrate tissue. This results in the production of high linear energy transfer (LET), α-particles (^4^He), and recoiling lithium-7 (^7^Li) nuclei [1]. Because of the short trajectory of these heavy particles (5–9 μm; the distance is approximately less than a single cell diameter), radiation damage is limited to those targeted (cancer) cells containing ^10^B. Thus, if ^10^B compounds can selectively target only cancer cells, side effects typically associated with ionizing radiation can be avoided [2].

For successful BNCT treatment, a critical ^10^B amount and sufficient energy from thermal neutrons must be delivered to individual cancer cells in the tumor lesions. Ideal B compounds should exhibit the following characteristics: low systemic cytotoxicity; selective uptake by the tumor; high ratios of tumor/blood (T/Bl) and tumor/surrounding normal tissue (T/N) (preferably greater than 3–4:1), which minimizes the impact that the capture of neutrons by hydrogen and nitrogen, the two elements present in high concentrations in the body, can have on the absorbed radiation dose; 20–35 µg of ^10^B/g of tumor lesions, which accounts for 10^9^ atoms (numbers) of ^10^B/cell; rapid clearance from blood and normal tissues; persistence in the tumor for the duration of the treatment [3].

The first generation of developed B compounds was sodium borocaptate (BSH) and boronophenylalanine (BPA), which have been used against various types of cancer cells [4]. BPA, an amino acid analog, is a leading B compound for the clinical treatment (BNCT) of malignant melanomas. Research groups have reported that L-amino acid transporter-1 (LAT-1) preferably imports BPA into malignant tumor cells [5,6]. BSH is a super-hydrophilic anionic B compound that has 12 ^10^B atoms per molecule, but it is hydrophilic and has no specific uptake mechanisms for any tumor cells. It relies on the blood–brain barrier to achieve selective accumulation in tumors relative to the brain [6,7,8]. Despite encouraging results achieved in clinical trials worldwide, these compounds feature low tumor selectivity or low B content per molecule [9].

Following the development of BNCT for cancer treatment, advancing the medicinal chemistry of polyhedral B clusters rather than those with a single B atom per molecule has been pursued. Among the new compounds discovered, “carborane”, highly hydrophobic and comprising two carbon (C) atoms and ten B atoms with three isomers, is characterized by the substituents of *ortho (o) (1, 2), meta (m) (1, 7),* and *para (p) (1, 12)* in the occupied position of their diagram [10].

In the early stage of BNCT, natural B atoms consisting of 19.9% ^10^B and 80.1% ^11^B are used, while in the clinical stages, the ^10^B-enriched carriers are used because of their high neutron capture cross-section of 3837 barns as compared to ^11^B with 0.005 barns [11,12]. Recent efforts to improve the selectivity of B compounds have involved incorporating them into tumor-targeting molecules such as peptides, proteins, antibodies, nucleosides, sugars, porphyrins, liposomes, and nanoparticles [13]. One strategy for improving the delivery of B compounds to tumors is using polymeric nanoparticles as a drug delivery system (DDS), which has attracted much attention in terms of controlled release and targeting [9,14].

The recent research has explored a hydrophobic B cluster conjugated with a tumor-targeting molecule such as *o-*carborane-loaded poly(L-lactide-*co*-glycolide) nanoparticles [15], cyclic RGD-functionalized *closo-*dodecarborate albumin [16], and sulfonamide-functionalized carborane [17].

One example of biologically self-degradable polymers that have high potential for tumor imaging and antitumor therapy is the “AB-type Lactosome’’ nanoparticle (AB-Lac particle). The particles are composed of amphipathic polydepsipeptide, which is linked with a hydrophilic polysarcosine (PSar) and a hydrophobic poly-L-lactic acid (PLLA), and this polymer assembly forms micelle-type particles with an average diameter of 36 nm [14,18]. Moreover, the AB-Lac particles show promising prospects for solid tumor accumulation via the enhanced permeability and retention (EPR) effect, which allows particles in the size range 10–100 nm to leak from blood vessels having submicron-sized defects in the tumor lesions where the lymphatic drain system is generally immature [19,20].

In the present study, we aimed to develop AB-Lac particles highly loaded with a hydrophobic B compound that can be applied in the BNCT treatment. As candidates, we selected three carborane isomers (*o*-carborane, *m*-carborane, and *p*-carborane), and three *o*-carborane derivatives (1, 2-dimethyl-*o*-carborane (diC1-Carb), 1, 2-dihexyl-*o*-carborane (diC6-Carb), and 1, 2-didodecyl-*o*-carborane (diC12-Carb)). We unexpectedly discovered the unique interaction between the AB-Lac particles and one of carborane derivatives, i.e., diC6-Carb, namely, a “molecular glue” effect. It was originally reported as inducing the formation of protein–protein interactions to elicit biologic or therapeutic effects and has been viewed enthusiastically as a unique pharmacological modality to target proteins without degradable pockets, although the interactions were discovered serendipitously [21]. The molecular glue effect was also found in bifunctional biologics such as bispecific antibodies and by the existence of stabilizers [22], the configuration between optically active biodegradable polymers PLLA and PDLA [23], and nanofibers [24]. Based on these findings, the molecular glue effect in AB-Lac particles loaded with diC6-Carb could occur because the AB-Lac particles have a polydepsipeptide structure and stick by six-alkyl chains in diC6-Carb [25,26].

To confirm the efficacy of the hydrophobic B compound-loaded AB-Lac particles, a set of in vitro, in vivo, and ex vivo studies was planned. For assessing accumulation of the hydrophobic B compound-loaded AB-Lac particles in tumor lesions and other organs, we performed in vivo and ex vivo near-infrared fluorescence (NIRF) imaging and inductively coupled plasma atomic emission spectroscopy (ICP-AES) analysis in murine breast cancer cell (4T1)-derived xenografts. The optimum accumulation of the particles in the xenografts was assessed using NIRF imaging with the fluorescence agent Indocyanine Green labeled PLLA_34_ (ICG-PLLA_34_), which was also composed into the B compound-loaded AB-Lac particles. We suggested that a particular hydrophobic alkylated *o*-carborane derivative (i.e., diC6-Carb)-loaded AB-Lac particle represents a promising strategy as a novel BNCT medicine that deploys enough quantities of ^10^B into tumor tissues (lesions), even if we have not yet attempted to evaluate it by any neutron irradiation trials.

## 2. Materials and Methods

### 2.1. Reagents

All reagents were purchased commercially and were of reagent grade. The AB-Lac polymer, PSar_106_-block-PLLA_32_ (molecular weight: 10,001) (Figure 1A), was synthesized by KNC Laboratories, Co., Ltd. (Kobe, Japan). *o*-Carborane, *m*-carborane, *p*-carborane, diC1-Carb, diC6-Carb, and diC12-Carb (Figure 1B) were purchased from Katchem, Ltd. (Prague, Czech Republic). ^10^B-BPA was purchased from Interpharma Praha, a.s. (IPP) (Prague, Czech Republic). RPMI-1640 medium, Dulbecco’s modified phosphate buffered saline (DPBS), chloroform, *N*, *N*-dimethylformamide (DMF), and nitric acid (HNO_3_) were purchased from Fujifilm Wako Pure Chemical Corp. (Osaka, Japan). Dimethyl sulfoxide (DMSO) and fetal bovine serum (FBS) were purchased from Sigma-Aldrich, Co., Ltd. (St. Louis, MO, USA). B standard solution, penicillin–streptomycin mixed solution, and the trypsin (2.5 g/L)-EDTA (1 mmol/L) solution were purchased from Nacalai Tesque, Inc. (Kyoto, Japan). The Cell Counting Kit-8 (CCK-8) was purchased from Dojindo Molecular Technologies, Inc. (Kumamoto, Japan).

### 2.2. Cell Culture

Cell lines, i.e., murine breast cancer (4T1), murine colorectal carcinoma (CT26), human pancreatic cancer (AsPC-1), and human gastric carcinoma (NCI-N87), were obtained from the American Type Culture Collection (ATCC) (Rockville, MD, USA), and they were incubated in RPMI-1640 medium supplemented with 10% (*v*/*v*) of FBS and 1% (*v*/*v*) of the penicillin/streptomycin solution in an incubator under 5% CO_2_ atmosphere at 37 °C. These cells were subcultured using the trypsin (2.5 g/L)-EDTA (1 mmol/L) solution for cell detachment.

### 2.3. Preparation of the B Compound-Loaded AB-Lac Particles

The mixture of PSar_106_-block-PLLA_32_ in chloroform (1 mL, 1 µmol) and 100 µL of B cluster (10 µmol) in chloroform (or DMF) were added to a test tube and evaporated to form a polymer-B compound film. Then, 2.0 mL of DPBS was added and treated with a bath-type sonicator for 20 min, followed by conditioning the mixture with the PD-10 desalting column (GE Healthcare, Buckinghamshire, UK) and the nanoparticle-enriched eluate was collected. The eluate was then passed through a 0.22 µm syringe filter (Merck Millipore, Dublin, Ireland), followed by filtration at 0.1 µm (PALL Corporation, NY, USA) to exclude larger-sized and aggregated particles (Figure 1C). The B compounds loaded into the AB-Lac particles comprised *o-*carborane, *m-*carborane, *p-*carborane, diC1-Carb, diC6-Carb, or diC12-Carb. The loaded B amount in the particles was measured with ICP-AES (Shimadzu, Kyoto, Japan). The particle size distribution (PSD) and polydispersity index (PdI) of the AB-Lac particles were determined by taking 40 µL of sample solution into a disposable low volume cuvette (GmbH & Co. KG, Weinheim, Germany) and a Zetasizer (Nano ZSP; Malvern, Instruments, Malvern, UK) for 60 s equilibration time. The PSD and PdI data are represented as mean ± S.D.

### 2.4. Preparation of the ICG-Labeled AB-Lac Particles

A mixture of PSar_106_-*block*-PLLA_32_ in chloroform (1 mL, 1 µmol) and 1 mol% of ICG-PLLA_34_ in chloroform was evaporated to form a film in a test tube. A B compound (10 µmol) dissolved in 100 µL of chloroform (or DMSO) was added into the same test tube and stored for 30 min until the polymer film completely dissolved. After evaporating, the mixture was resuspended in DPBS (1.9 mL) at 1200 rpm stirring for 1 h, followed by conditioning the mixture with the PD-10 desalting column, and the nanoparticle-enriched eluate was collected. The eluate was then passed through the 0.22 and 0.1 µm syringe filters. The optical density (OD) of the particle suspensions (at 794 nm) was measured with the BioSpec spectrometer (Shimadzu). The B amount was also measured with ICP-AES.

### 2.5. Stability Study: Releasing Test of B from B Cluster-Loaded AB-Lac Particles

The leakage of carborane isomers or *o*-carborane alkylated derivatives from the loaded AB-Lac particles was evaluated by a dialysis test. In brief, the 2 mL aliquot was poured into a dialysis tubing cellulose membrane (20/32; MWCO: 12,000–14,000; Nacalai Tesque, Inc, Kyoto, Japan), and each tube was immersed in 300 mL of DPBS or DPBS with 10% FBS at 4 or 37 °C. The 2 mL of the solution outside the dialysis tube was taken at 0, 1, 3, 6, 12, 24, 48, or 72 h and the same volume of the corresponding DPBS was filled back. ICP-AES measurement was conducted to determine the B amount of carborane isomers or *o*-carborane alkylated derivatives released from the loaded AB-Lac particles. PSD and PdI of the loaded particles inside the dialysis tube were compared as “before” and “after” the dialysis. The B amount data are represented as mean ± S.E.M while PSD and PdI are represented as mean ± S.D.

### 2.6. Cell Cytotoxicity

The cancer cells (4T1, 5 × 10^3^ cells/well) were seeded into a 96-well microplate and incubated under 5% CO_2_ at 37 °C for 24 h prior to the treatment. Subsequently, the cells were treated with *o*-carborane, *m-*carborane, diC1-Carb, or diC6-Carb-loaded AB-Lac particles at concentrations of 25, 50, 100, and 250 ppm as the B amount equivalent for 24 h. The quadruplicate assays (*n* = 4) were conducted for the cell cytotoxicity test using the CCK-8 according to the kit’s instructions. The OD was measured at 450 nm using a microplate reader (Sunrise™, Tecan Trading AG, Manne Dorf, Switzerland). The cell viability was calculated as % by comparison with the non-treated control group. Data are represented as mean ± S.E.M.

### 2.7. In Vitro Cell Uptake of B Cluster-Loaded AB-Lac Particles: Time- and Dose-Dependent Studies

The cancer cells, i.e., 4T1 (0.5 × 10^6^/mL), CT26 (0.5 × 10^6^/mL), AsPC-1 (1.0 × 10^6^/mL), or NCI-N87 (1.0 × 10^6^/mL), were seeded in a 6-well plate and incubated under 5% CO_2_ at 37 °C for 24 h prior to the treatment. Subsequently, the cells were treated with *o-*carborane, *m-*carborane, diC1-Carb, or diC6-Carb-loaded AB-Lac particles (at 0.15, 0.5, or 2 mM of B equivalent) for 2, 4, or 6 h (*n* = 3). After the treatment, the cells were washed 3 times with DPBS, harvested by the trypsin treatment, and collected by centrifugation at 1800 rpm for 5 min. The cell pellets were digested using 1 mL of 60% HNO_3_ overnight at room temperature. After filtering with a 0.50 µm filter (Advantech, 13JP050AN, Toyo Roshi, Ltd., Tokyo, Japan), purified water was added up to 10 mL before the ICP-AES analysis. Data are represented as mean ± S.E.M. Significant differences were represented by ** *p* < 0.05 and * *p* < 0.10.

### 2.8. Xenograft-Tumor Model

Six-week-old female nude mice (BALB/c nu/nu) were purchased from Charles River (Yokohama, Japan). Two weeks before the study, the 4T1 cells (5 × 10^6^ cells) suspended in 100 µL of DPBS were subcutaneously (*s.c.*) inoculated in the right thigh of the mice. The weight of the mice and the tumor volume were monitored every 2–3 days. All animal experiments were approved by the Animal Care and Use Committee of Okayama University (OKU-2021448). 

### 2.9. In Vivo and Ex Vivo NIRF Imaging

The ICG-labeled AB-Lac particles (100 µL) or those loaded with B compounds were *i.v.* injected via the tail of the respective xenografts (4 mice in each group). The in vivo NIRF imaging was taken at 1, 3, 6, 12, 24, 48, or 72 h after the injection. After the last 72 h imaging, the mice were euthanized. Then, the organs, i.e., blood, heart, lung, liver, spleen, pancreas, kidney, stomach, intestine, muscle, and tumor, were excised and weighed. The ex vivo NIRF imaging of each organ was taken to determine the fluorescence intensity using the IVIS spectrum system (Xenogen, Hopkinton, MA, USA) with specific filters for ICG (excitation at 745 nm and emission at 840 nm).

### 2.10. Ex Vivo B Biodistribution of B-Cluster-Loaded AB-Lac Particles

The AB-Lac particles loaded with *o*-carborane or those with diC6-Carb (at respective B doses of 5 mg B/kg) were *i.v.* injected via the tail of respective 4T1-tumor-bearing mice (4 mice in each group). The mice were euthanized, and the organs, i.e., blood, heart, lung, liver, spleen, pancreas, kidney, stomach, intestine, muscle, and tumor, were excised and weighed at 24 h post-injection. The excised organs were digested with HClO_4_ and H_2_O_2_ (1:1) at 90 °C for 2 h, and then, the digested samples were diluted with 10 mL of purified water. After filtering with a 0.50 µm filter, the samples were measured with ICP-AES analysis. Data are represented as mean ± S.E.M. Significant differences were represented by ** *p* < 0.05 and * *p* < 0.10.

## 3. Results

### 3.1. Selection of B Compounds-Loaded AB-Lac Particles

Six kinds of B compounds were individually loaded into the AB-Lac particles: *o*-carborane, *m*-carborane, *p*-carborane, diC1-Carb, diC6-Carb, or diC12-Carb. These were carborane isomers and/or hydrophobic *o-*carborane derivatives with different alkyl substitutions. These B compounds were dissolved in two different solvents (chloroform and DMF) to be loaded into the AB-Lac particles. The results showed that those loaded had the highest B:AB-Lac polymer ratio (the ratio (B:AB-Lac polymer)) of 10.0 when *o*-carborane was dissolved in chloroform to form the particles (Table 1). In contrast, the lowest ratio (0.23) was obtained with the *m-*carborane dissolved in DMF. As shown in Table 1, the loaded amount of the isomers varied and depended on their solubility in a solvent.

In general, hydrophobicity of the alkylated derivative of *o*-carborane increased depending on the total length of two alkyl chains [27]. Moreover, the highest ratio (2.7) was obtained with diC6-Carb dissolved in chloroform rather than with diC1-Carb or diC12-Carb.

In addition to the ratio (B:AB-Lac polymer) of the particles loaded with *o*-carborane or its related compounds, their PSD was assessed (Figure 2). The results indicated that all three isomers dissolved in chloroform had a similar PSD of around 60-88 nm after passing through the PD-10 column, followed by two filtration processes. It was also shown that uncommon PSD with *p-*carborane (dissolved in DMF) formed into two peaks; one was relatively smaller (49 ± 1.1 nm) than a regular size distribution of the AB-Lac particles but had an additional peak that possibly formed self-aggregated and larger-sized particles in the preparation processes, especially in DMF as a solvent. Moreover, when dissolving *m-*carborane in DMF, the PSD was relatively bigger, 88 ± 0.86 as compared to dissolving it in chloroform, 72 ± 1.5.

As shown in Table 1, the ratios (B:AB-Lac polymer) of AB-Lac particles loaded with one of the alkylated *o-*carborane derivatives, i.e., diC1-Carb, diC6-Carb, or diC12-Carb, were also calculated. Overall, the highest ratio of 2.7 was obtained in those loaded with diC6-Carb (by dissolving in chloroform). The ratios for diC1-Carb and diC12-Carb were 1.2 and 0.6, respectively. The PSD of *o-*carborane derivatives was measured, and the results are also shown in Figure 2. The particle size (based on PSD data) of AB-Lac particles loaded with diC1-Carb (dissolved in chloroform) was 64 ± 0.93 nm, similar to those with carborane isomers. In contrast, the AB-Lac particles loaded with diC6-Carb revealed a significantly large size (PSD: 100 ± 0.40 nm) as compared to those with any of the three *o*-carborane isomers, and the PSD with diC12-Carb was a moderate size, i.e., 93 ± 0.74 nm.

To confirm the possibility that diC12-Carb formed particles by themselves, we featured each preparing process for the AB-Lac particles loaded with diC12-Carb (Appendix A). The results revealed a polydisperse pattern after passing through the PD-10 column without two-step filtrations, and large-sized aggregates remained. The results also confirmed that diC12-Carb may have formed larger particles (aggregates) by themselves after only a small amount of the B compound was incorporated in the AB-Lac particles.

In all cases of *o-*carborane compounds, except the *p*-carborane particles, B compound-loaded AB-Lac particles (after passing through the PD-10 column and the 2-step filtrations) showed near monodispersity, with PdI around 0.21–0.32 (Figure 2 and Appendix A).

### 3.2. Stability Study: Releasing Test of B from the B Compound-Loaded AB-Lac Particles

The PSD of the plain AB-Lac particles (vehicle) and those loaded with *o*-carborane, *m*-carborane, diC1-Carb, or diC6-Carb was determined before and after 24 h dialysis against DPBS at 4 °C or 37 °C and against DPBS with 10% FBS at 37 °C (Figure 3). Overall, all data on the AB-Lac particles loaded with any carborane compounds (except with diC6-Carb) indicated that the particle size partly/mostly shifted to become another larger-sized particle ( > 80 nm) and revealed a polydisperse pattern after dialysis in the presence or absence of 10% FBS at 37 °C, but not at 4 °C. Thus, the data indicated the “temperature-dependent instability” of the PLLA α-helix core in so-called “biodegradable” nanoparticles. On AB-Lac particles loaded with *m*-carborane (PSD = 110 ± 0.12 against DPBS and PSD = 92 ± 0.90 against DPBS with 10% FBS) and loaded with diC1-Carb (PSD = 120 ± 1.5 against DPBS and PSD = 82 ± 0.90 against DPBS with 10% FBS) intermediate sizes alternatively appeared with dialysis at 37 °C, and diC1-Carb was sustainable.

However, the AB-Lac particles loaded with diC6-Carb showed an entirely different trend, and a slightly larger particle size of around 92-110 nm was maintained even at 37 °C. From the PSD data, those loaded with diC6-Carb also showed consistent intensity before/after dialysis owing to the tight interaction between the PLLA α-helix and two hexyl chains of diC6-Carb by, namely, the “molecular glue” effect.

To assess the stability of the B compound-loaded AB-Lac particles in another way, we observed the cumulative B release from the AB-Lac particles loaded with *o*-carborane, *m*-carborane, diC1-Carb, or diC6-Carb (Figure 4). The AB-Lac particles loaded with diC1-Carb had the highest cumulative B release at 37 °C in around 71% against DPBS and 96% against DPBS and 10% FBS after 24 h while showing significantly lower cumulative B release at 4 °C in around 45%. Subsequently, the cumulative B release of those loaded with *o-*carborane was 46% against DPBS and 80% against DPBS with 10% FBS at 37 °C, which was higher than dialysis at 4 °C with 32% after 24 h. Furthermore, the cumulative B release of those with *m-*carborane showed a low and similar trend in both dialysis conditions against DPBS, at 4 and 37 °C, with 12% and 13%, respectively, while against DPBS with 10% FBS it was significantly high at around 73%. diC6-Carb showed a completed limited (no) B leakage in both conditions where dialysis at 37 °C had 0% of cumulative B release at 24 h.

### 3.3. Cell Cytotoxicity of the B-Compound Loaded AB-Lac Particles

The study was performed to confirm the cytotoxicity of the B compound-loaded AB-Lac particles at different B concentrations of 25, 50, 100, and 250 ppm in in vitro 4T1 cancer cell culture. After 24 h incubation, the CCK-8 reagent was added to each well. The results showed no cytotoxicity at 250 ppm, even at 2 mM (216 ppm of B equivalent), which was an adjusted concentration for the cell uptake study (Appendix A).

### 3.4. In Vitro Cell Uptake Study of AB-Lac Particles Loaded with a B Compound: Time- and Dose-Dependent Studies

A time-dependent in vitro cell uptake study of the AB-Lac particles loaded with *o-*carborane and of BPA (B: 2 mM equivalent) was performed in 4T1 cells (Appendix A). Overall, the results showed that cell uptake of the B amount from the AB-Lac particles loaded with *o-*carborane was higher than that of BPA at any tested time points. Time-dependency of cell uptake of the AB-Lac particles loaded with *o-*carborane showed the highest uptake with 13 × 10^9^ of B number/cell at 2 h, while the 4 and 6 h uptakes were 8.6 and 12 × 10^9^ of B number/cell, respectively. The cell uptake of the B atoms of the AB-Lac particles loaded with *o-*carborane was higher than those of BPA, i.e., 4.6, 4.6, and 9.5 × 10^9^ of B number/cell at the respective times. On the graph, the primary *y*-axis (the left side) indicates B number (1 × 10^9^)/cell, and the secondary *y*-axis (the right side) indicates ng B per 10^6^ cells.

Subsequently, in vitro cell uptake of the AB-Lac particles loaded with *o-*carborane and with diC6-Carb was evaluated with four different cancer cell lines, which were two murine cancer cell lines (4T1 and CT26) and two human cancer cell lines (AsPC-1 and NCI-N87). The respective cells were treated for 2 h with the particles loaded with *o-*carborane and with diC6-Carb, respectively, at B concentrations equivalent to 0.15, 0.5, and 2 mM.

Overall, the dose-dependent effect showed in all four cancer cell lines by incubating 0.5 mM of the B equivalent of the AB-Lac particles loaded with diC6-Carb, and the B concentration was sufficient to meet the minimum BNCT requirement of cell uptake, i.e., 1 × 10^9^ B atoms/cell (equivalent to ca. 20 ng B/10^6^ cells). It also indicated that the AB-Lac particles loaded with diC6-Carb had a higher cell uptake than those loaded with *o-*carborane (Figure 5).

In the comparative cell uptake studies of the murine cancer cell line and human cancer cell line, the murine cancer cell line showed a higher B uptake than the human cancer cell line, with the highest B uptake in 4T1 incubated in AB-Lac loaded with diC6-Carb (at 2 mM B concentration equivalent) with 35 × 10^9^ B number/cell.

The cells loaded with *o-*carborane reached the highest B uptake in CT26 with 19 × 10^9^ B number/cell. Moreover, when comparing both human cancer cell lines, AsPC-1 and NCI-N87, those loaded with diC6-Carb had similar B uptakes while those loaded with *o-*carborane had a higher B uptake in NCI-N87. Regarding the cell uptake mechanism, those loaded with *o-*carborane and loaded with diC6-Carb were likely internalized into the cancer cell line by endocytosis. On the graph, the primary *y*-axis (the left side) indicates in B number (×10^9^)/cell and the secondary *y*-axis (the right side) indicates in ng B/10^6^ cells.

### 3.5. In Vivo and Ex Vivo NIRF Imaging: Biodistribution of the AB-Lac Particles

The biodistribution of particles in tumor lesions was observed in the AB-Lac particles labeled with ICG-PLLA_34_. The prepared ICG-labeled AB-Lac particles (100 µL) were *i.v.* injected via the tail of the xenografts (four mice in each group). After that, in vivo NIRF imaging was taken at different time points from 1 to 72 h post-injection to observe the optimum accumulation time of ICG-labeled AB-Lac particles in tumor lesions.

The result showed that the accumulation of ICG-labeled AB-Lac particles gradually accumulated in the tumor lesions, probably owing to the EPR effect, and reached the peak at 12 h post-injection, which was slightly higher than at 24 h as indicated by the fluorescence intensity. Moreover, as the time elapsed, the ratio of tumor/normal tissue increased and reached the peak at 72 h post-injection (Figure 6). These data indicated that the ICG (as a NIRF probe)-labeled AB-Lac particles and concealed in the hydrophobic core spread over the whole body through the blood circulation at around 1-3 h after injection, and the intensity in normal regions decreased as time elapsed to 72 h.

After 72 h, the mice were euthanized, and the blood, heart, lung, liver, spleen, pancreas, kidney, stomach, intestine, muscle, and tumor were excised, followed by the quantification of ex vivo NIRF imaging (Figure 7). The imaging result showed that the tumor tissues had the strongest intensity as compared to other organs, with the second highest intensity found in the liver.

### 3.6. Biodistribution of the AB-Lac Particles Loaded with a B Compound

The B biodistribution in the organs of the xenograft mice was analyzed. The AB-Lac particles loaded with *o-*carborane or diC6-Carb (5 mg of B equivalent/kg) were *i.v.* injected via the tail. After 24 and 72 h, the mice were euthanized, and the organs were excised followed by digestion in HClO_4_:H_2_O_2_ (1:1), and the amount of B in each organ was measured by ICP-AES (Figure 8).

At 24 h post-injection, the data (Figure 8A) indicated that the AB-Lac particles loaded with diC6-Carb, 13 µg/g, had a higher B accumulation in the tumor, as compared to those with o-carborane, 11 µg/g. The AB-Lac particles loaded with *o*-carborane showed remarkable accumulation in the liver, stomach, and intestine, while the AB-Lac particles loaded with diC6-Carb showed high accumulation in the stomach and intestine. At 72 h, the data (Figure 8B) indicated that the B amount significantly decreased as compared to 24 h post-injection for AB-Lac particles loaded with either diC6-Carb or *o*-carborane, at 1.2 and 0.94 µg/g, respectively. Furthermore, the B accumulation in each organ declined significantly as compared to 24 h post-injection. The T/N ratio in the AB-Lac particles loaded with diC6-Carb was higher than that with *o-*carborane after either 24 or 72 h post-injection (Figure 8C). After the 72 h post-injection, the T/N ratio in AB-Lac particles loaded with *o-*carborane decreased to 1.2:1, as compared to 2.3:1 at 24 h post-injection. However, the T/N ratio in AB-Lac particles loaded with diC6-Carb remained stable with 2.8:1 at 24 h and 2.9:1 at 72 h.

Furthermore, the T/Bl ratio indicated a similar pattern compared to the T/N ratio in which the AB-Lac particles loaded with diC6-Carb were higher than those with *o-*carborane (Figure 8D). The T/Bl ratio in the AB-Lac particles loaded with *o-*carborane was 3.1:1 at 24 h, then declined at 72 h with a ratio of 2.2:1, while the T/Bl ratio in those loaded with diC6-Carb almost doubled at 72 h as compared to that at 24 h, with a ratio of 5.5:1.

## 4. Discussion

Carboranes are the most widely studied of the B clustered compounds for medicinal chemistry [11]. Carborane possesses unique physicochemical properties, such as being highly hydrophobic and consisting of 10 B atoms [28]. To promote a BNCT role, the DDS delivering carborane derivatives has been explored because carborane derivatives bound to nanoparticles have advantages that include high stability and high accumulation in cancer cells [29].

We incorporated carborane isomers (*o*-carborane, *m*-carborane, and *p*-carborane), and *o*-carborane derivatives (diC1-Carb, diC6-Carb, and diC12-Carb) into the AB-Lac particles. Carborane isomers are well-known as highly hydrophobic B clusters with two carbon atoms located at the *ortho (o)-, meta (m)-,* or *para (p)*-position. The relative position of the two carbons in an isomer may influence their hydrophobicity. *p-*Carborane with 1, 12-isomers of carbon substitution is the most hydrophobic, followed by *m-*carborane with 1, 7-isomers and *o-*carborane with 1, 2-isomers [30,31]. We also applied three kinds of *o*-carborane derivatives, which included two carbons substituting for double alkyl chains of methyl (CH_3_), hexyl (C_6_H_13_), or dodecyl (C_12_H_25_). Their hydrophobicity is dependent on the total length of alkyl chain(s) in general, and a compound having longer alkyl chain(s) is more hydrophobic than those with shorter ones [27]. Thus, the hydrophobic order of *o-*carborane derivatives is diC12-Carb > diC6-Carb > diC1-Carb.

First, we considered what kind of solvent was suitable to dissolve the B compounds to prepare the B compound-loaded AB-Lac particles. Chloroform is a relatively less polar solvent with relative polarity of 0.26, while DMF has a higher polarity of 0.39 [32]. Based on the yield, it was shown that a low-polarity solvent such as chloroform was suitable rather than DMF to dissolve all three carborane isomers. *o-*carborane was highly soluble in either DMF or chloroform in our study. Detailed chemical features of these carborane isomers were reviewed elsewhere [33,34,35].

Among all six tested carborane-related B compounds, *o*-carborane was most yielded into the AB-Lac particles by dissolving in either of the tested solvents (Table 1). The high incorporation was probably due to hydrogen bonding between *o*-carborane and the PLLA core of the AB-Lac polymer [34]. *m-*Carborane also had sufficient yield by incorporation into the AB-Lac particles.

Subsequently, a stability study of the AB-Lac particles was performed (Figure 4). First, the stability study of the AB-Lac particles (without any carborane compound) showed that after dialysis under physiological conditions of 37 °C without/with 10% FBS, double peaks appeared. These phenomena confirmed the AB-Lac particles as self-biodegradable polymers that swelled when the degradation was significantly influenced by pH and temperature [36].

Furthermore, from both the B compound-incorporating study and the stability study of the AB-Lac particles, we summarized the interaction of the B compounds with the PLLA core (Figure 9). First of all, *o-*carborane had the highest ratio (B:AB-Lac polymer) but the B release rate from the loaded AB-Lac particles was also significantly high under certain physiological conditions, i.e., at 37 °C. The *o-*carborane may have a weak and insufficient interaction with the AB-Lac particles so that it can allow B to leak from the particles [37]. This can be explained by the conclusion that a nucleophilic reaction structurally converts the “*closo*”-form of *o-*carborane into the more hydrophilic and shortened size of another form, namely, the “*nido*”-form [38]. The *nido*-form can be passed through the pores of the dialysis membrane (cut-off: < 12,000–14,000 kDa). An interesting observation appears in Figure 3 and Figure 4, where the *m*-carborane-loaded particles of intermediate size (PSD = 110 ± 0.12) alternatively appearing by dialysis at 37 °C were sustainable and unable to leak out through the dialysis membrane in DPBS, although significant B release was shown in DPBS with 10% FBS.

As described above, the third isomer, *p-*carborane, was the most hydrophobic carborane isomer. Because of this feature, the resulting particles revealed their polydispersity, and the larger-sized particles as well as those of diC12-Carb may have been self-formed ones (aggregates) (Appendix A). A previous publication [39] reported that *p-*carborane had the highest pKa value (26.8) by the substituent CH, as compared to the other two isomers, followed by those of *m-*carborane (25.6) and of *o-*carborane (22.0). Therefore, in another study, the order of hydrogen bond strength was expected to be *o-*carborane > *m-*carborane > *p-*carborane [40], which was consistent with the order of the capsulated amount of carborane isomers dissolved in chloroform (Table 1). Moreover, among the three isomers, the *o*-carborane was the most reactive owing to higher inductive electron attraction (−1) than *m*-carborane and *p*-carborane, easier to functionalize, and more prone to undergo deboronation that changed this form from extremely hydrophobic *closo*-carboranes to *nido*-carboranes [10,41]. Furthermore, in *o-*carborane derivatives, the deboronation reaction was strongly influenced by the effect of the substituents. Specifically, if the substituents were categorized as electron-withdrawing substituents that easily electronically deprived the C-atom of carborane, the reactivity to the *nido-*form seemed to be increased [42,43]. As reflected in the alkyl chain in *o-*carborane derivatives, the alkyl chain is an electron-donating substituent where the reactivity to the *nido-*form is decreased. We expect that the reactivity order is *o-*carborane > diC1-Carb > diC6-Carb > diC12-carb. In the case of diC6-Carb, a bulkier substituent with greater steric hindrance impeded such a nucleophilic attack in an initial deboronation reaction.

The most notable observation was that one of the *o-*carborane derivatives (i.e., diC6-Carb) had a unique effect, the so-called “molecular glue” effect that can provide sustainable stability of the loaded AB-Lac particles even at 37 °C against either DPBS or DPBS with 10% FBS. In other observed carborane isomers and *o-*carborane derivatives, dialysis against DPBS with 10% FBS had a higher cumulative B release as compared to DPBS at 37 °C, but in diC6-Carb, the B release remained the same at 0% in both conditions. The force may be generated by the interaction between two hexyl (C_6_H_13_) chains at the *o-*position of carborane and the outer surface of the PLLA helix structures, owing to the highly hydrophobic interaction in 6.2 Å (0.62 nm) [44,45]. This was originally reported regarding the effect of inducing the formation of protein–protein interactions to elicit biologic or therapeutic effects and has been viewed enthusiastically as a unique pharmacological modality to target proteins without degradable pockets, although the interactions were discovered serendipitously [21].

The molecular glue effect explaining the two PLLA helix interaction patterns with their enantiomer PDLA was previously reported by Tsuji et al. on active polymers (L-configured or D-configured polymer) where unsubstituted/substituted optically active poly(lactic acid) can act as “a configurational/helical molecular glue” for two oppositely configured, optically active polymers (two D-configured polymers or two L-configured polymers) to allow for their co-crystallization [23]. However, such molecules are rare and have been discovered fortuitously, thus limiting their potential as a general strategy for therapeutic intervention [46].

Compared to the instability of the AB-Lac particle with o-carborane at 37 °C, the two methyl (CH_3_) chains in the diC1-Carb structure appeared to have no additional effect on the α-helix–α-helix interaction in the PLLA core, owing to the sufficient length of the alkyl chains. In contrast, DiC12-Carb had the longest alkyl chains, dodecyl (C_12_H_25_), which made it highly hydrophobic, so that it was revealed to self-form larger sized-particles (aggregates). Moreover, a small amount of diC12-Carb was also detected in the AB-Lac particles. Thus, we could conclude that diC12-Carb had a limited interaction with the PLLA core (Appendix A). Regarding the most hydrophobic carborane isomer, i.e., *p*-carborane, it may form self-aggregated and larger-sized particles in the preparation processes, especially in DMF as a solvent, as well as diC12-Carb (Figure 2D).

The in vitro cell uptake study also showed that the AB-Lac particles loaded with diC6-Carb were highly incorporated as compared to those with *o-*carborane for all four tested cancer cell lines (Figure 6). Those loaded with *o-*carborane were considered unstable with B significantly leaking out within 3–6 h, as shown in the stability study (Figure 4). Therefore, we selected 2 h incubation because a small amount, around 20% of B, was released from the nanoparticles for the AB-Lac particles loaded with *o-*carborane. Otherwise, diC6-Carb could be stably packed into the AB-Lac particles by the molecular glue effect. The evidence was also supported by the time-dependent in vitro cell uptake (Appendix A). AB-Lac particles loaded with *o-*carborane did not show a time-dependent effect, while BPA showed a time-dependent effect at 2–6 h incubation.

The time-dependent in vitro cell uptake (Appendix A) implied that the AB-Lac particles loaded with *o-*carborane were likely taken up by cancer cell lines through “endocytosis” because previous studies have also reported that the selective uptake of certain kinds of nanoparticles and hydrophobic cancer drugs internalized by the targeted cancer cell line were mediated by “endocytosis” [6,47]. Furthermore, a previous report indicated that several nanoparticles were rapidly taken up by the cells through an endocytosis mechanism, phagocytosis, within 15 min in gold nanorods [48]. Conversely, hydrophilic BPA is mainly taken up by the cancer cell line via the L-amino acid transporter-1 (LAT-1), which is overexpressed in cancer cells, with a small amount also being transported by the LAT-2, which is overexpressed in both cancer cells and normal cells [49].

In previous studies [50,51], different types of NIRF-labeled AB-Lac particles were injected into tumor-bearing mice. They observed a long duration of fluorescence intensity, i.e., the ratio (tumor:liver) continued to increase after 24-48 h, and the NIRF-labeled nanoparticles remained in the blood circulation to accumulate at the tumor site owing to the EPR effect, similar to the results in the present study. Our ex vivo NIRF imaging showed that the tumor had the strongest intensity as compared to other organs, with the second highest intensity found in the liver. The evidence led us to consider that the major excretion was via the liver. Furthermore, the AB-Lac particles were retained in the tumor sites at 72 h post-injection, probably owing to the EPR effect but not entirely.

In contrast, the ex vivo B biodistribution at 24 h indicated that the B amounts in the tumor lesions exceeded the minimum requirement for a therapeutic effect in BNCT. The minimum requirement was expected to be around 15–20 µg/g [52], although a significant decrease was observed at 72 h post-injection in those loaded with *o-*carborane and diC6-Carb. The remarkable B amounts also showed in the liver, stomach, and intestine, which indicated the major excretion via enterohepatic pathway, such as in the ex vivo NIRF imaging. This significant accumulation occurred because nanoparticles tend to be rapidly cleared out via vascular, renal, and enterohepatic pathways. This tendency occurred on difference nanoparticles, such as dendrimers, inorganic biodegradable co-polymers, quantum dots, carbon nanoparticles, liposomes, and silica, as previously reported by Longmire et. al. [53]. Therefore, the B amount in various organs and in blood circulations significantly decreased at 72 h post-injection. The AB-Lac particles loaded with diC6-Carb indicated that the ratios of T/N and T/Bl remained at around 3:1 at both 24 and 72 h, and the T/Bl ratio even increased at 72 h post-injection. However, the T/N and T/Bl ratios in the AB-Lac particles loaded with *o-*carborane significantly decreased at 72 h, as compared to those at 24 h post-injection, and the requirement for T/N and T/Bl ratios for BNCT is ideally more than 3:1 [54].

Thus, the B biodistribution study confirmed the “molecular glue” effects on the AB-Lac particles loaded only with diC6-Carb, which provided a good stability with the T/N and T/Bl ratios, and further extensive study will be performed to strengthen the evidence.

In this study, we developed AB-Lac particles that showed promise as a DDS for BNCT with a remarkable EPR effect. At the same time, we observed other types of amphipathic polymer DDSs for BNCT, namely, “A_3_B type Lactosome (A_3_B-Lac)” composed of three hydrophilic PSar and a hydrophobic PLLA. Lim, M et al. reported that the A_3_B-Lac conjugated with small interfering RNA (siRNA), having a relatively lower EPR effect on tumor lesions down to 22 nm [19,55]. To overcome the disadvantages and enhance the penetration in tumors, we developed a 27 kDa human single chain variable fragment (scFv) of IgG against the mesothelin (MSLN) overexpressed in various cancer cells [56]. Our future study must elucidate whether the anti-MSLN-scFv-labeled A_3_B-Lac particles loaded with diC6-Carb are more suitable as a BNCT medicine.

## 5. Conclusions

In summary, diC6-Carb proved to be a promising B delivery agent that was highly incorporated in the AB-Lac particles, as compared to the other tested B compounds. An in vitro study demonstrated that the B cellular uptake met the requirements whereby the B compound of the AB-type Lactosome (AB-Lac) particles loaded with diC6-Carb accumulated ≥ 10^10^ atoms of ^10^B per cell (under a non-cytotoxic dose) to cancer cells. Moreover, in vivo and ex vivo studies showed that the EPR effect occurred with the B compound-loaded AB-Lac particles, reaching the peak of intensity at 12 h post-injection, and imaging of the excised tumors and organs indicated that tumors had the highest intensity in comparison with other organs. The in vivo B biodistribution study indicated that a sufficient B amount accumulated in the tumor, and the T/N and T/Bl ratios remained stable from 3 until 72 h post-injection. Hence, these current data suggested that further optimization and investigations are required to develop a suitable diC6-Carb-loaded nanoparticle as a promising candidate for BNCT.

## Figures and Tables

**Figure 1 cells-11-03307-f001:**
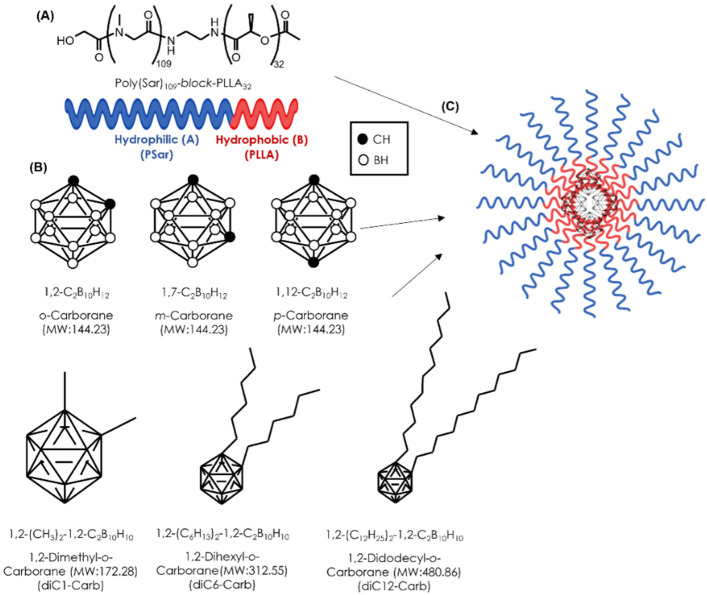
The structural concept of a novel DDS for BNCT. The structure of the AB-Lac particles (**A**); the chemical structure of hydrophobic B compounds (**B**); the schematic illustration of the AB-Lac particles loaded with a hydrophobic B compound (**C**).

**Figure 2 cells-11-03307-f002:**
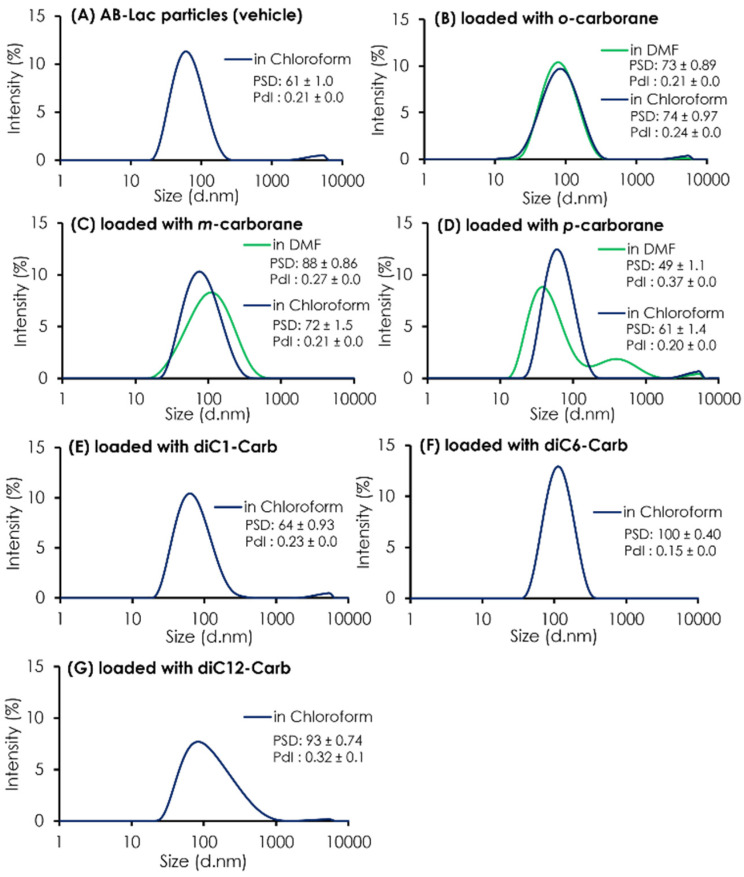
The PSD and PdI of AB-Lac particles loaded with a B compound. The AB-Lac particles (plain vehicles) (**A**), and those containing *o*-carborane (**B**), *m-*carborane (**C**), *p-*carborane (**D**), diC1-Carb (**E**), diC6-Carb (**F**), and diC12-Carb (**G**). The PSD and PdI are represented as mean ± S.D. (*n* = 3).

**Figure 3 cells-11-03307-f003:**
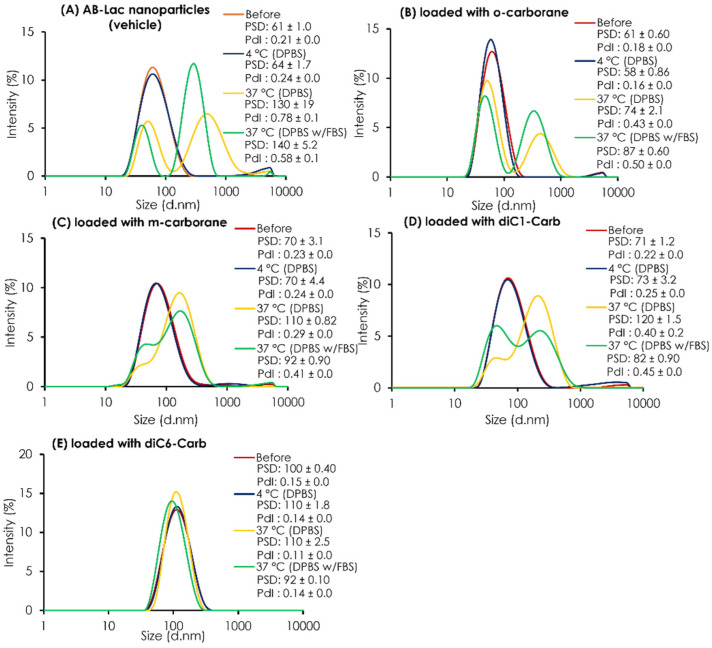
The stability test of AB-Lac particles loaded with B compounds. The PSD and PdI inside a dialysis bag before and after 24 h dialysis. The AB-Lac particles (vehicle) (**A**), loaded with *o-*carborane (**B**), with *m-*carborane (**C**), with diC1-Carb (**D**), and with diC6-Carb (**E**) against DPBS at 4 °C and either against DPBS or DPBS with 10% FBS at 37 °C were calculated. The PSD and PdI are represented as mean ± S.D. (*n* = 3).

**Figure 4 cells-11-03307-f004:**
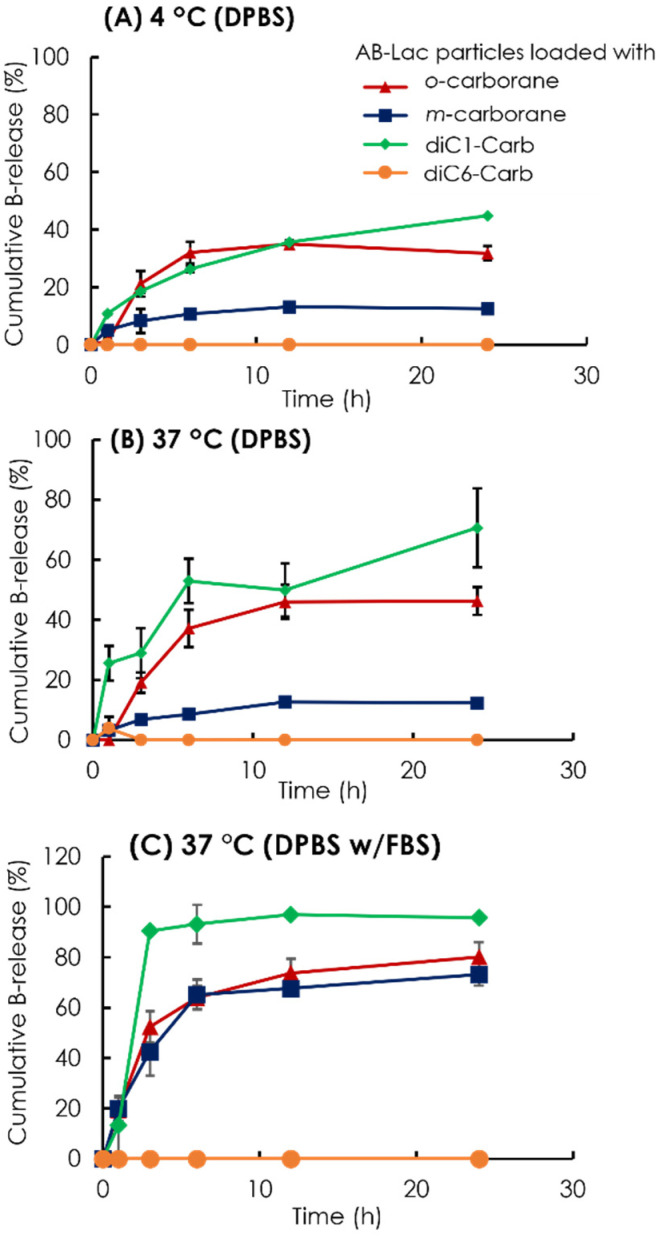
Time-dependent % leakage of B from the AB-Lac particles loaded with *o*-carborane, *m*-carborane, diC1-Carb, and diC6-Car by dialysis at 4 °C (**A**) and 37 °C (**B**) against DPBS and 37 °C against DPBS with 10% FBS (**C**). The B release was measured by taking the solution outside the dialysis bag at respective time points using ICP-AES. Data are expressed as mean ± S.E.M. (*n* = 3).

**Figure 5 cells-11-03307-f005:**
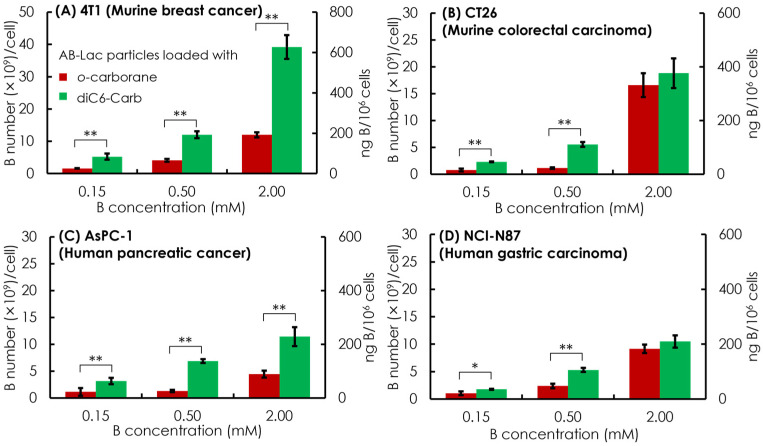
In vitro cell uptake of the AB-Lac particles loaded with *o-*carborane or diC6-Carb in different cancer cells, 4T1 (**A**), CT26 (**B**), AsPC-1 (**C**), and NCI-N87 (**D**). Data are expressed as mean ± S.E.M. (*n* = 3). Significant differences are represented by ** *p* < 0.05 and * *p* < 0.10.

**Figure 6 cells-11-03307-f006:**
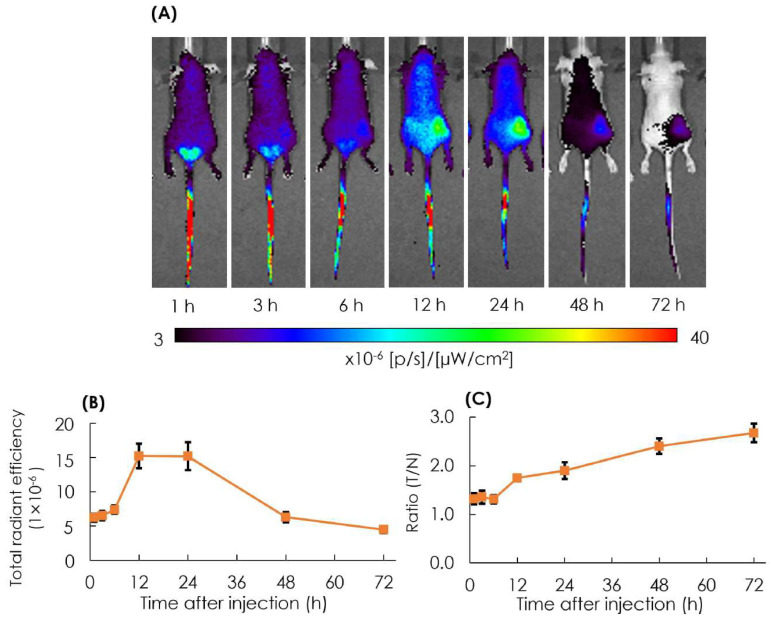
In vivo NIRF imaging of the 4T1 tumor-bearing mice after *i.v.* injection with ICG-labeled AB-Lac particles (**A**); the graph of fluorescence intensity in tumor lesions (**B**) and the ratio of tumor intensity in normal tissue injected with ICG-labeled AB-Lac particles (**C**). Data are expressed as mean ± S.E.M. (*n* = 4).

**Figure 7 cells-11-03307-f007:**
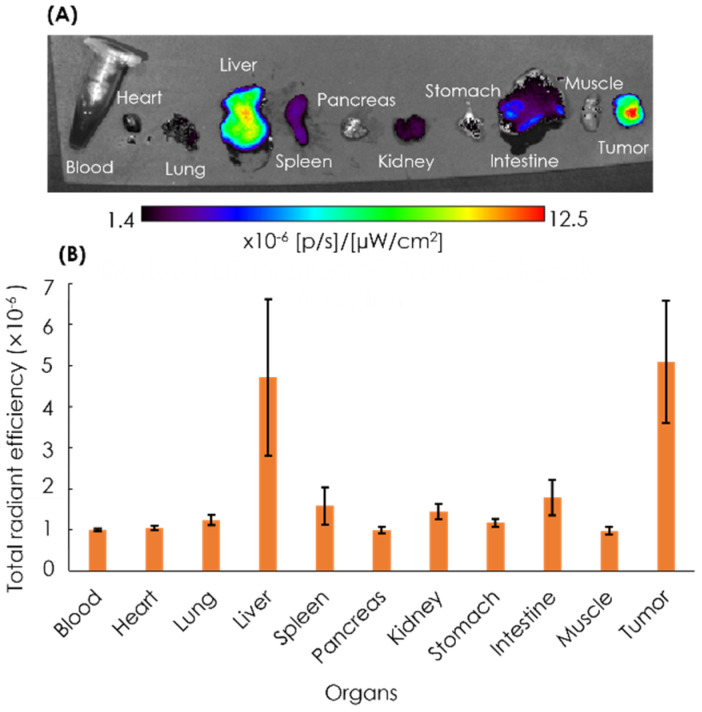
Ex vivo NIRF imaging of organs (**A**) and the quantification of ICG fluorescence (**B**). Ex vivo NIRF imaging of excised organs of the xenografts was performed at 72 h post-injection with ICG-labeled AB-Lac particles. Data are expressed as mean ± S.E.M. (*n* = 4).

**Figure 8 cells-11-03307-f008:**
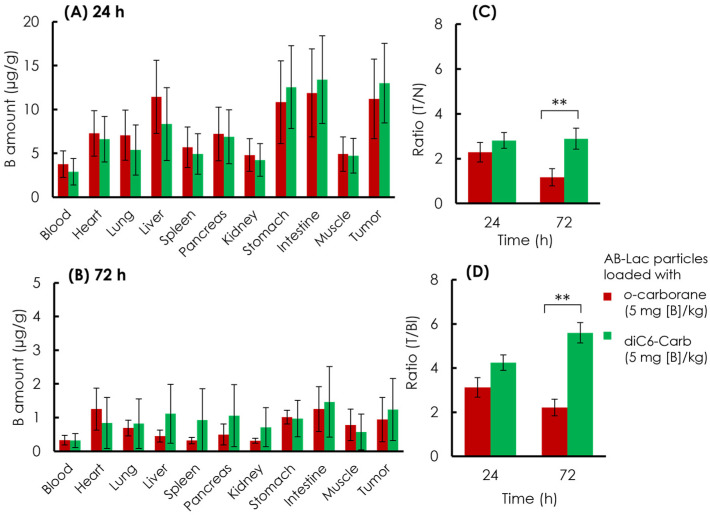
Ex vivo B biodistribution of the AB-Lac particles loaded with B compounds. B amount in tumor lesions at 24 h (**A**) and 72 h (**B**); T/N ratio (**C**) and (**D**) T/Bl ratio at 24 and 72 h. Data are expressed as mean ± S.E.M. (*n* = 4). Significant differences are represented by ** *p* < 0.05.

**Figure 9 cells-11-03307-f009:**
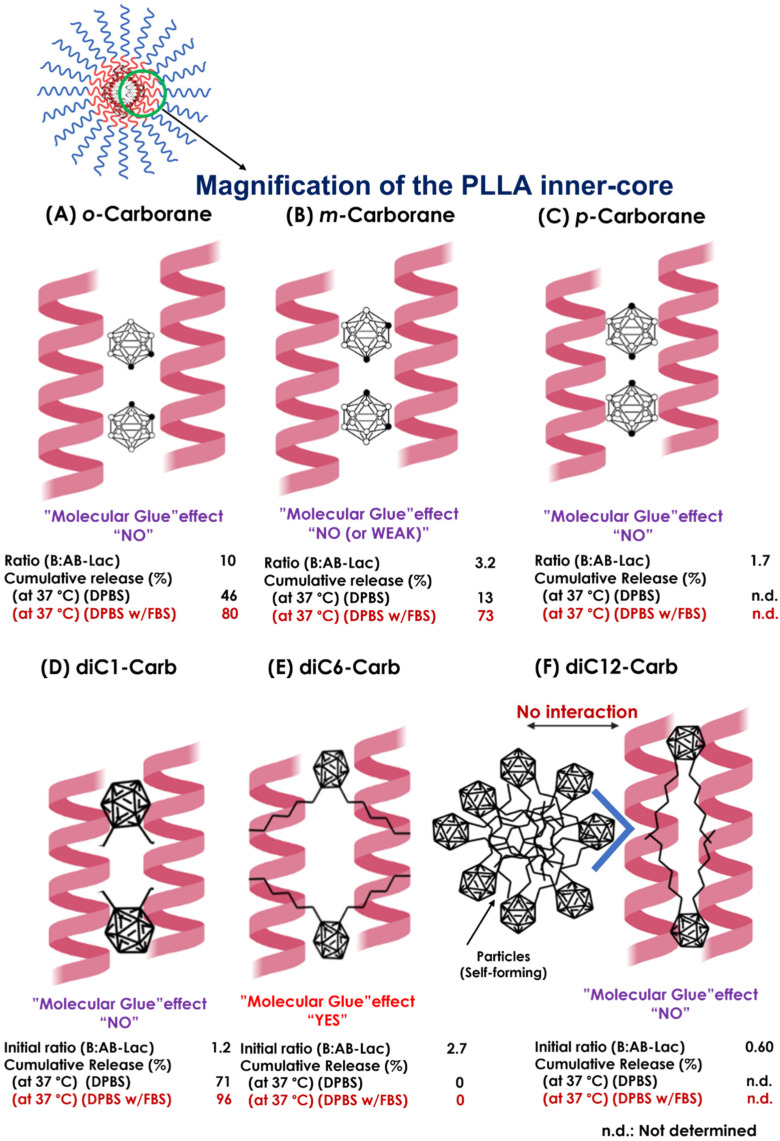
The schematic representations of presumable interactions between the B compound and PLLA core helix in the AB-Lac particles. Possible interactions of the PLLA core with o-carborane (**A**), m-carborane (**B**), p-carborane (**C**), diC1-Carb (**D**), diC6-Carb (**E**), and diC12-Carb (**F**). The ratio (B:AB-Lac polymer) refers to Table 1.

**Table 1 cells-11-03307-t001:** The B compounds loaded in the AB-Lac particles.

Carborane Isomer/ *o-*Carborane Derivative	In Initial Preparation		In the Final Particles
	Carborane (µmol)	AB-Lac (µmol)	The Ratio (B:AB-Lac Polymer) (µmol:µmol)
AB-Lac particles loaded with			
*o-*Carborane	* 10	1.0	5.8
	10	1.0	10
*m-*Carborane	* 10	1.0	0.23
	10	1.0	3.2
*p-*Carborane	* 10	1.0	2.7
	10	1.0	1.7
diC1-Carb	10	1.0	1.2
diC6-Carb	10	1.0	2.7
diC12-Carb	10	1.0	0.60

* Dissolved in DMF; remainder of cases: dissolved in chloroform. The ratio (B:AB-Lac polymer) was calculated with the “determined B amount” in the forming particles and the “estimated amount” of AB-Lac polymer in the initial preparation.

## Data Availability

Not applicable.

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
