# Peer review of "Novel Self-Forming Nanosized DDS Particles for BNCT: Utilizing A Hydrophobic Boron Cluster and Its Molecular Glue Effect"

_cells, 2022, doi:10.3390/cells11203307_

Round 1
Reviewer 1 Report
Authors propose nanoparticles formed of alkylated o-carborane and PSar106-block-PLLA32 copolymer (AB-Lac) as potential boron carrier for BNCT. It is a worthwhile work and discovered exceptionally strong hydrophobic interaction between PLLA α-helix of AB-Lac and two hexyl chains of diC6-Carb (perhaps with participation of the boron cluster) is intriguing. Albeit, using the term “molecular glue” coined for small molecule that sticks two proteins together to describe the observed by authors phenomenon, is far-fetched.
After a preliminary study and comparison of the properties of three carborane isomers and three o-carborane derivatives alkylated on carbon atoms with alkyl groups of different lengths, authors selected diC6-Carb for further research. The extensive and detailed studies of nanoparticles formed with diC6-Carb loaded Ab-Lac shown that the obtained nanoparticles may fulfil the main requirements for boron carriers for BNCT justyfying their further studies.
However, as for some properties, unmodified o-carborane do not give way to the complicated system of AB-Lac/diC6-Carb, e.g cellular uptake for o-carborane and AB-Lac/diC6-Carb is similar in CT26 and NCI-N87 cells at higher concentration (Fig. 5), uptake in tumor tissue and T/N ratio after 24 h are quite similar (Fig. 8) for both, o-carborane and AB-Lac/diC6-Carb. These requires authors’ comment and consideration.
In addition:
1) The procedure for PSD measurements should be added into Experimental part.
2) The time-dependent release of boron containing compounds from the AB-Lac particles Fig. 4 should be reconsidered. Are the observed differences in release of born congaing compounds due to stability of the boron compound AB-Lack nanoparticle or due to resistance of carborane derivative to closo- to nido-carborane transformation increasing the cluster/cluster derivative solubility? How the solubility/hydrophobicity could affect diffusion through the membrane of the dialysis tubing? It is also worth taking into account that the influence of the methyl group and longer alkyl substituents on the electronic properties of the cluster and, consequently, the susceptibility to closo / nido transformations is different.
Also, do the observed effect of temperature on boron compound release from the AB-Lac particles is due to lower stability of boron compound loaded AB-Lac nanoparticles at higher temperature or rather due to faster diffusion of released boron compound through a membrane at higher temperature?
Although it is obvious that the authors focus on studies relevant to BNCT, the lack of in-depth analysis and research on the observed unexpected affinity of diC6-Carb towards Ab-Lac is unsatisfied. The sketches shown in Fig. 9 will not replace systematic physical chemistry, molecular modeling and quantum-mechanical studies. It seems that the topic is worth such effort.
In conclusion, I recommend publishing this work after addressing the concerns listed above and after a very thorough and professional correction of the English language.
Author Response
Reviewer 1
- Authors propose nanoparticles formed of alkylated o-carborane and PSar106-block-PLLA32 copolymer (AB-Lac) as potential boron carrier for BNCT. It is a worthwhile work and discovered exceptionally strong hydrophobic interaction between PLLA α-helix of AB-Lac and two hexyl chains of diC6-Carb (perhaps with participation of the boron cluster) is intriguing. Albeit, using the term “molecular glue” coined for small molecule that sticks two proteins together to describe the observed by authors phenomenon, is far-fetched.
Thank you for the comments, we made additional description in page 3 (lines 105 -110) that the AB-Lac particles had a polydepsipeptide structure and the previous published papers revealed a “molecular glue effect” are additionally cited.
“The molecular glue effect was also found in bifunctional biologics such as bispecific antibodies and by the existence of stabilizers [22], the configuration between optically active biodegradable polymers PLLA and PDLA [23], and nanofibers [24]. Based on these findings, the molecular glue effect in AB-Lac particles loaded with diC6-Carb could occur because the AB-Lac particles have a polydepsipeptide structure and stick by six-alkyl chains in diC6-Carb [25,26].”
We also described the molecular glue effect that occurred on optically active biodegradable polymer, PLLA and PDLA in page 15 (line 518 – 523).
“The molecular glue effect explaining the two PLLA helix interaction patterns with their enantiomer PDLA was previously reported by Tsuji et al. on active polymers (L-configured or D-configured polymer) where unsubstituted/substituted optically active poly(lactic acid) can act as “a configurational/helical molecular glue” for two oppositely configured, optically active polymers (two D-configured polymers or two L-configured polymers) to allow their co-crystallization [23].”
Additional references are listed:
- Geiger, T.M.; Schäfer, S.C.; Dreizler, J.K.; Walz, M.; Hausch, F. Clues to molecular glues. Current Research in Chemical Biology 2022, 2, 100018, doi:10.1016/j.crchbi.2021.100018.
- Tsuji, H.; Noda, S.; Kimura, T.; Sobue, T.; Arakawa, Y. Configurational molecular glue: One optically active polymer attracts two oppositely configured optically active polymers. Sci Rep 2017, 7, doi:10.1038/srep45170.
- Lee, E.; Kim, J.-K.; Lee, M. Lateral Association of Cylindrical Nanofibers into Flat Ribbons Triggered by “Molecular Glue.” Angewandte Chemie International Edition 2008, 47, 6375–6378, doi:10.1002/anie.200801496.
- Hara, E.; Ueda, M.; Makino, A.; Hara, I.; Ozeki, E.; Kimura, S. Factors Influencing in Vivo Disposition of Polymeric Micelles on Multiple Administrations. ACS Med Chem Lett 2014, 5, 873–877, doi:10.1021/ml500112u.
- Uji, H.; Watabe, N.; Komi, T.; Sakaguchi, T.; Akamatsu, R.; Mihara, K.; Kimura, S. Downsizing to 25-nm reverse polymeric micelle composed of AB3-type polydepsipeptide with comprising siRNA. Chem Lett 2022, 51, 235–238, doi:10.1246/cl.210704.
- After a preliminary study and comparison of the properties of three carborane isomers and three o-carborane derivatives alkylated on carbon atoms with alkyl groups of different lengths, authors selected diC6-Carbfor further research. The extensive and detailed studies of nanoparticles formed with diC6-Carb loaded Ab-Lac shown that the obtained nanoparticles may fulfil the main requirements for boron carriers for BNCT justifying their further studies.
Thank you for the suggestions. We described the planned of further study in the last paragraph of discussion in page 17 (lines 582 – 594).
“Thus, the B biodistribution study confirmed the “molecular glue” effects on the AB-Lac particles loaded only with diC6-Carb, which provided a good stability with the T/N and T/Bl ratios, and further extensive study will be performed to strengthen the evidence. In this study, we developed AB-Lac particles that showed promise as a DDS for BNCT with a remarkable EPR effect. At the same time, we observed other types of amphipathic polymer DDSs for BNCT, namely, “A3B type Lactosome (A3B-Lac)” composed of three hydrophilic PSar and a hydrophobic PLLA. Lim, M et al. reported that the A3B-Lac conjugated with small interfering RNA (siRNA), having a relatively lower EPR effect on tumor lesions down to 22 nm [19,55]. To overcome the disadvantages and enhance the penetration in tumors, we developed a 27 kDa human single chain variable fragment (scFv) of IgG against the mesothelin (MSLN) overexpressed in various cancer cells [56]. Our future study must elucidate whether the anti-MSLN-scFv-labeled A3B-Lac particles loaded with diC6-Carb are more suitable as a BNCT medicine.”
- However, as for some properties, unmodified o-carborane do not give way to the complicated system of AB-Lac/diC6-Carb, e.g. cellular uptake for o-carborane and AB-Lac/diC6-Carb is similar in CT26 and NCI-N87 cells at higher concentration (Fig. 5), uptake in tumor tissue and T/N ratio after 24 h are quite similar (Fig. 8) for both, o-carborane and AB-Lac/diC6-Carb. These requires authors’ comment and consideration.
The uptake of AB-Lac loaded with carborane compound by cancer cells were so rapidly and reached to plateau by 2 hour-incubation in vitro. The uptake was also dose-dependent of these particles (as B equivalent). In the in vitro cell culture condition (i.e., 37°C at high dose of B equivalent (2 mM)), quietly similar uptake was observed with o-carborane and diC6-Carb. However, at lower dose of B equivalent (0.15 and 0.50 mM), the AB-Lac particles loaded with diC6-Carb was preferably uptaken by that of o-carborane. In vivo and ex vivo kinetic studies, the rapid uptake of those particles into tumor lesions was also observed. At 24 hr-post injection of the particles, relatively high amount of particles are present in various organs and in the blood circulation. However, those are metabolized to disappear after 24 hrs up to 72 hrs and that resulted in increasing not only T/N ratio but also T/Bl ratio, especially in a group of AB-Lac loaded with diC6-Carb as compared to those with o-carborane. Thus, the molecular glue effect may contribute sustaining of the diC6-Carb loaded particles in the tumor lesions.
We described our consideration and comments for in vitro study in page 16 (line 536-555)
“The in vitro cell uptake study also showed that the AB-Lac particles loaded with diC6-Carb were highly incorporated as compared to those with o-carborane for all four tested cancer cell lines (Figure 6). Those loaded with o-carborane were considered unstable with B significantly leaking out within 3-6 h, as shown in the stability study (Figure 4). Therefore, we selected 2 h incubation because a small amount, around 20% of B, was released from the nanoparticles for the AB-Lac particles loaded with o-carborane. Otherwise, diC6-Carb could be stably packed into the AB-Lac particles by the molecular glue effect. The evidence was also supported by the time-dependent in vitro cell uptake (Figure S3). AB-Lac particles loaded with o-carborane did not show a time-dependent effect, while BPA showed a time-dependent effect at 2-6 h incubation.
The time-dependent in vitro cell uptake (Figure S3) implied that the AB-Lac particles loaded with o-carborane were likely taken up by cancer cell lines through “endocytosis” because previous studies have also reported that the selective uptake of certain kinds of nanoparticles and hydrophobic cancer drugs internalized by the targeted cancer cell line were mediated by “endocytosis” [6,47]. Furthermore, a previous report indicated that several nanoparticles were rapidly taken up by the cells through an endocytosis mechanism, phagocytosis, within 15 min in gold nanorods [48]. Conversely, hydrophilic BPA is mainly taken up by the cancer cell line via the L-amino acid transporter-1 (LAT-1), which is overexpressed in cancer cells with a small amount also being transported by the LAT-2, which is overexpressed in both cancer cells and normal cells [49].”
For in vivo study in page 16 – 17 (line 571 – 581)
“This significant accumulation occurred because nanoparticles tend to be rapidly cleared out via vascular, renal, and enterohepatic pathways. This tendency occurred on difference nanoparticles, such as dendrimers, inorganic biodegradable co-polymers, quantum dots, carbon nanoparticles, liposomes, and silica, as previously reported by Longmire et al [53]. Therefore, the B amount in various organs and in blood circulations significantly decreased at 72 h post-injection. Interestingly, the AB-Lac particles loaded with diC6-Carb indicated that the ratios of T/N and T/Bl remained at around 3:1 at both 24 and 72 h, and the T/Bl ratio even increased at 72 h post-injection. However, the T/N and T/Bl ratios in the AB-Lac particles loaded with o-carborane significantly decreased at 72 h, as compared to those at 24 h post-injection, and the requirement for T/N and T/Bl ratios for BNCT is ideally more than 3:1 [54].”
We also added statistical difference in both Figure 5 (page 5) and figure 8 (page 13) that represented by **p < 0.05 and *p < 0.10.
- The procedure for PSD measurements should be added into Experimental part.
Thank you for the suggestion, we already added the procedure of PSD and PdI measurements in page 4 (lines 161 – 165)
“The particle size distribution (PSD) and polydispersity index (PdI) of the AB-Lac particles were determined by taking 40 µL of sample solution into a disposable low volume cuvette (GmbH & Co. KG, Weinheim, Germany) and a Zetasizer (Nano ZSP; Malvern, Instruments, Malvern, UK) for 60 s equilibration time. The PSD and PdI data are represented as mean ± S.D.“
- The time-dependent release of boron containing compounds from the AB-Lac particles Fig. 4 should be reconsidered. Are the observed differences in release of born congaing compounds due to stability of the boron compound AB-Lack nanoparticle or due to resistance of carborane derivative to closo- to nido-carborane transformation increasing the cluster/cluster derivative solubility? How the solubility/hydrophobicity could affect diffusion through the membrane of the dialysis tubing? It is also worth taking into account that the influence of the methyl group and longer alkyl substituents on the electronic properties of the cluster and, consequently, the susceptibility to closo / nido transformations is different. Also, do the observed effect of temperature on boron compound release from the AB-Lac particles is due to lower stability of boron compound loaded AB-Lac nanoparticles at higher temperature or rather due to faster diffusion of released boron compound through a membrane at higher temperature?
Thank you for the questions, first of all, the AB-Lac particles are consist of a self-biodegradable polymer that can be swelled under a physiological condition and their sizes are also influenced by pH and temperature as written in page 14 (lines 461 – 465). We considered at higher temperature, due to nanoparticles was swelled also influence boron release.
“First, the stability study of the AB-Lac particles (without any carborane compound) showed that after dialysis under physiological conditions of 37 °C without/with 10% FBS, double peaks appeared. These phenomena confirmed the AB-Lac particles as self-biodegradable polymers that swelled when the degradation was significantly influenced by pH and temperature [36].
Reference
- da Silva, D.; Kaduri, M.; Poley, M.; Adir, O.; Krinsky, N.; Shainsky-Roitman, J.; Schroeder, A. Biocompatibility, biodegradation and excretion of polylactic acid (PLA) in medical implants and theranostic systems. Chemical Engineering Journal 2018, 340, 9–14, doi:10.1016/j.cej.2018.01.010.
We also described that among the isomers, o-carborane is the most reactive and easily change the form from hydrophobic closo- to hydrophilic nido- in page 14 (lines 488 – 492). Due to hydrophilicity, the boron compounds were leaking out of the dialysis membrane
“Moreover, among the three isomers the o-carborane was the most reactive owing to higher inductive electron attraction (-1) than m-carborane and p-carborane, easier to functionalize, and more prone to undergo deboronation that changed this form from extremely hydrophobic closo-carboranes to nido-carboranes [10,41].”
References
- Stockmann, P.; Gozzi, M.; Kuhnert, R.; Sárosi, M.B.; Hey-Hawkins, E. New keys for old locks: Carborane-containing drugs as platforms for mechanism-based therapies. Chem Soc Rev 2019, 48, 3497–3512, doi:10.1039/c9cs00197b.
- Emilia O, Z.; Christian A, M.; Mark W Lee, J. The Use of carboranes in cancer drug development. Int J Cancer Clin Res 2019, 6, doi:10.23937/2378-3419/1410110.
Regarding the effect of alkylation, we considered longer alkylation affects on deboronation reaction that change closo- to nido- with the reactivity orders are o-carborane > diC1-Carb > diC6-Carb > diC12-carb. We explained in page 14 (lines 492 – 500).
“Furthermore, in o-carborane derivatives, the deboronation reaction was strongly influenced by the effect of the substituents. Specifically, if the substituents were categorized as electron-withdrawing substituents that easily electronically deprived the C-atom of carborane, the reactivity to the nido-form seemed to be increased [42,43]. As reflected in the alkyl chain in o-carborane derivatives, the alkyl chain is an electron-donating substituent where the reactivity to the nido-form is decreased. We expect that the reactivity order is o-carborane > diC1-Carb > diC6-Carb > diC12-carb. In the case of diC6-Carb and diC12-Carb, a bulkier substituent with greater steric hindrance impeded such a nucleophilic attack in an initial deboronation reaction.”
Reference
- Scholz, M.; Hey-Hawkins, E. Carbaboranes as Pharmacophores: Properties, Synthesis, and Application Strategies. Chem Rev 2011, 111, 7035–7062, doi:10.1021/cr200038x.
- Powell, C.L.; Schulze, M.; Black, S.J.; Thompson, A.S.; Threadgill, M.D. Closo → nido cage degradation of 1-(substituted-phenyl)-1,2-dicarbadodecaborane(12)s in wet DMSO under neutral conditions. Tetrahedron Lett 2007, 48, 1251–1254, doi:10.1016/j.tetlet.2006.12.034.
- Although it is obvious that the authors focus on studies relevant to BNCT, the lack of in-depth analysis and research on the observed unexpected affinity of diC6-Carb towards Ab-Lac is unsatisfied. The sketches shown in Fig. 9 will not replace systematic physical chemistry, molecular modeling and quantum-mechanical studies. It seems that the topic is worth such effort.
Thank you for the comments, we illustrated the figures based on our data we obtained. We realize that extensive studies needed to know exact interaction between the AB-Lac particles with o-carborane derivative and isomers. We also changed the caption in figure 9 (page 15).
“Figure 9. The schematic representations of presumable interactions between the B compound and PLLA core helix in the AB-Lac particles. Possible interactions of the PLLA core with o-carborane (A), m-carborane (B), p-carborane (C), diC1-Carb (D), diC6-Carb (E), and diC12-Carb (F). The ratio (B:AB-Lac polymer) refers to Table 1.”
- In conclusion, I recommend publishing this work after addressing the concerns listed above and after a very thorough and professional correction of the English language.
Thank you for the suggestion, in this latest version, English language already reviewed extensively by a MDPI’s professional English editor.

Reviewer 2 Report
There are several comments.
1. the experiments to determine stability are artificial. Stability was studied in DPBS, but in blood there are a large number of proteins that have both hydrophobic sites and specific binding centers with hydrophobic molecules (e.g., albumin). Therefore, the results obtained can be very different from the behavior of the obtained nanoparticles in biological fluids.
2. the biodistribution of AB-Lac nanoparticles was performed using NIRF imaging. A different method is used for nanoparticles containing boron. The authors write. After 24 and 72 hrs, the mice were euthanized, and the organs were excised followed by digestion in HClO4:H2O2 (1:1) and the B amount of each organ was measured by the ICP-AES (Figure 8). (lines 393-395). However, Figure 8 lacks information on the boron content of the organs, making it impossible to compare information on the biodistribution of AB-Lac nanoparticles and AB-Lac nanoparticles containing boron.
On what basis do the authors exclude the absence of accumulation of a large amount of boron, for example, in the liver?
Author Response
Reviewer 2
- The experiments to determine stability are artificial. Stability was studied in DPBS, but in blood there are a large number of proteins that have both hydrophobic sites and specific binding centers with hydrophobic molecules (e.g., albumin). Therefore, the results obtained can be very different from the behavior of the obtained nanoparticles in biological fluids.
Thank you for the suggestion. We added the data by showing the results of stability study against DPBS with 10% FBS. The data appeared in several place in the method, results and discussion part. We showed the new data of PSD and PdI before and after dialysis in Figure 3 (page 8) and cumulative boron release in Figure 4, panel C (page 9). In the boron release it was indicated that in against DPBS with 10% FBS the boron release was higher. We also indicated the dialysis against DPBS with 10%FBS in the data summary in Figure 9 (page 15).
Figure 3. The stability test of AB-Lac particles loaded with B compounds. The PSD and PdI inside a dialysis bag before and after 24 h-dialysis. The AB-Lac particles (vehicle) (A), loaded with o-carborane (B), with m-carborane (C), with diC1-Carb (D), and with diC6-Carb (E) against DPBS at 4 °C and either against DPBS or DPBS with 10% FBS at 37 °C were calculated. The PSD and PdI are represented as mean ± S.D. (n=3).
Figure 4. Time-dependent % leakage of B from the AB-Lac particles loaded with o-carborane, m-carborane, diC1-Carb, and diC6-Car by dialysis at 4 °C (A) and 37 °C (B) against DPBS and 37 °C against DPBS with 10% FBS (C). The B release was measured by taking the solution outside the dialysis bag at respective time points using ICP-AES. Data are expressed as mean ± S.E.M. (n=3).
- The biodistribution of AB-Lac nanoparticles was performed using NIRF imaging. A different method is used for nanoparticles containing boron. The authors write. After 24 and 72 hrs, the mice were euthanized, and the organs were excised followed by digestion in HClO4:H2O2 (1:1) and the B amount of each organ was measured by the ICP-AES (Figure 8). (lines 393-395). However, Figure 8 lacks information on the boron content of the organs, making it impossible to compare information on the biodistribution of AB-Lac nanoparticles and AB-Lac nanoparticles containing boron. On what basis do the authors exclude the absence of accumulation of a large amount of boron, for example, in the liver?
Thank you for the valuable comments. We changed the Figure 8 (page 12) that indicated B accumulation in each organ at 24 h and 72 h post injection. We added some explanation in page 12 (lines 404 - 412)
“At 24 h post-injection, the data (Panel A) indicated that the AB-Lac particles loaded with diC6-Carb, 13 µg/g, had a higher B accumulation in the tumor, as compared to those with o-carborane, 11 µg/g. The AB-Lac particles loaded with o-carborane showed remarkable accumulation in the liver, stomach, and intestine, while the AB-Lac particles loaded with diC6-Carb showed high accumulation in the stomach and intestine. At 72 h, the data (Panel B) indicated that the B amount significantly decreased as compared to 24 h post-injection for AB-Lac particles loaded with either diC6-Carb or o-carborane, at 1.2 and 0.94 µg/g, respectively. Furthermore, the B accumulation in each organ declined significantly as compared to 24 h post-injection.”
and page 16 (lines 569 – 571)
“The remarkable B amounts also showed in the liver, stomach, and intestine, which indicated the major excretion via enterohepatic pathway, such as in the ex vivo NIRF imaging.”

Round 2
Reviewer 2 Report
The article can be published in the present form.